

# Co-scattering in micrOMEGAs:
# A case study for the singlet-triplet dark matter model

**Gaël Alguero[1,2], Geneviève Bélanger[2], Sabine Kraml[1] and Alexander Pukhov[3]**

**1** Univ. Grenoble Alpes, CNRS, Grenoble INP, LPSC-IN2P3, Grenoble, France
**2** LAPTh, CNRS, USMB, 9 Chemin de Bellevue, 74940 Annecy, France
**3** Skobeltsyn Inst. of Nuclear Physics, Moscow State Univ., Moscow 119992, Russia

## Abstract

In scenarios with very small dark matter (DM) couplings and small mass splittings between the DM and other dark sector particles, so-called "co-scattering" or "conversion-driven freeze-out" can be the dominant mechanism for DM production. We present the inclusion of this mechanism in MICROMEGAS together with a case study of the phenomenological implications in the fermionic singlet-triplet model. For the latter, we focus on the transition between co-annihilation and co-scattering processes. We observe that co-scattering is needed to describe the thermal behaviour of the DM for very small couplings, opening up a new region in the parameter space of the model. The triplet states are often long-lived in this region; we therefore also discuss LHC constraints from long-lived signatures obtained with SMODELS.



# 1   Introduction

In the standard dark matter (DM) paradigm, a single weakly interacting massive particle (WIMP) forms the DM, and the annihilation of DM into Standard Model (SM) particles determines the DM relic abundance through the freeze-out mechanism [1]. Typically weak couplings and DM masses near the weak scale are required for this. Motivated in part by the lack of conclusive evidence for such WIMPs despite the extensive astrophysical and colliders search program underway [2–5], recently a much larger range of DM masses and couplings have been explored, and different mechanisms for DM formation have been proposed.

In particular, weaker couplings of the DM ($\chi$) to SM particles can lead to a value for the DM relic density that is consistent with the one extracted from measurements of the cosmic microwave background [6]. This can occur in models where the dark sector[1] contains several new particles and co-annihilation processes involving heavier states of the dark sector (which we generically denote $\psi$) can set the scale for the relic density [7,8]. Co-annihilation of a dark particle with the DM ($\psi\chi \to \text{SM SM}$) or self-annihilation of two dark particles ($\psi\psi \to \text{SM SM}$) require a small mass splitting between DM and the heavier state(s) such that the number density of the $\psi$s is not too strongly Boltzmann suppressed. The states responsible for co-annihilation have at least weak couplings, while the coupling of the DM to the SM can be suppressed. Nonetheless, however, the DM is assumed to be in thermal equilibrium with the SM, for example through processes like $\chi \text{ SM} \to \chi \text{ SM}$.

Another possibility is DM co-scattering [9] or conversion-driven freeze-out [10], where inelastic scattering processes such as $\chi \text{ SM} \to \psi \text{ SM}$ are responsible for DM formation. This also requires small mass splitting between $\chi$ and $\psi$, but involves very small couplings between the DM and the particles in the thermal bath. In such scenarios, chemical equilibrium between $\chi$ and $\psi$ is not maintained and one needs to solve separate Boltzmann equations for $\chi$ and $\psi$, which are coupled through a conversion term involving co-scattering as well as (inverse) decay processes. For even smaller DM couplings, one enters the regime of the freeze-in mechanism, where DM is so feebly interacting that it is not in equilibrium with the SM in the early Universe [11,12].

The computation of annihilation and co-annihilation processes for DM freeze-out has long been standard in public DM tools such as MICROMEGAS [13], DARKSUSY [14], SUPERISORELIC [15] or MADDM [16], following the framework developed in [17,18]. The freeze-in mechanism was incorporated more recently in MICROMEGAS [19] and DARKSUSY [20], while co-scattering and decay processes have not yet been included in public tools. This is the gap that we start to fill with this work. Concretely, we present in this paper the implementation of the co-scattering mechanism[2] in MICROMEGAS together with a case study of the phenomenological implications in the singlet-triplet fermions model (STFM).

The STFM extends the SM with a singlet $\chi$ and a triplet $\psi$ of fermions, which are both odd under a new $\mathbb{Z}_2$ symmetry. In the context of supersymmetry, this has its equivalence in the bino-wino scenario of the minimal supersymmetric standard model (MSSM), i.e. $\chi \approx \tilde{B}$, $(\psi^\pm, \psi^0) \approx (\tilde{W}^\pm, \tilde{W}^0)$; in particular it is realised in Split Supersymmetry [21,22] with heavy higgsinos and a heavy gluino. It is generally a prime example of a model which can produce the dark matter relic density via either co-annihilation or co-scattering. Co-annihilation in the singlet-triplet (or bino-wino) model has been extensively discussed in the literature, see e.g. [23–29]. The same framework was also used to discuss the usage of the full momentum-dependent Boltzmann equations for co-scattering processes in [30]. On the collider side, the

---

[1]Here we define each dark sector to be made of all particles that possess the same symmetry properties, in particular under the discrete symmetry that stabilizes DM, and that are in thermal equilibrium with each other.

[2]When referring to co-scattering as a mechanism, we mean the inclusion of both inelastic scattering and (inverse) decays in the conversion terms of the Boltzmann equations.

model can lead to signatures of long-lived particles, which are heavily searched for at the LHC [31]. Depending on the mass splitting and on the size of the couplings involved, relevant signatures can include disappearing tracks and/or heavy stable charged particles.

The incorporation of co-scattering in MICROMEGAS relies heavily on the machinery developed to include multi-component DM as it requires to solve at least two separate Boltzmann equations, one for each set of dark particles in thermal equilibrium [32,33]. We do, however, make the simplifying assumption that DM is maintained in kinetic equilibrium and solve the fully integrated Boltzmann equations. This presents a limitation of the current work, since, when too weak couplings are involved, departure from kinetic equilibrium in the early Universe will impact the relic density calculation [34], see also [10, 30, 35]. This can lead to a sizeable systematic uncertainty on the computed $\Omega h^2$.

The paper is organised as follows. In Section 2, we present the STFM, which is used as the showcase model in this work. In Section 3, we discuss the relic density calculation for the co-scattering mechanism as incorporated in MICROMEGAS. Section 4 then contains a numerical analysis for the STFM; this includes a discussion of co-annihilation versus co-scattering, taking into account relic density as well as LHC constraints. Our conclusions are presented in Section 5. The new MICROMEGAS routines relevant for co-scattering are described in Appendix A.

## 2 Singlet-triplet extension of the SM

As mentioned in the Introduction, we will illustrate the physics case and phenomenological implications by means of the fermionic singlet-triplet model, STFM. This model extends the SM by two electroweak multiplets: a fermionic singlet $\chi$ and a fermionic SU(2)$_L$ triplet $\psi$, which are both odd under a new $\mathbb{Z}_2$ symmetry, while the SM particles are even. Following the notation of [30], but with four-component Majorana spinors, the most general Lagrangian for this model is

$$\mathcal{L} = \mathcal{L}_{\text{SM}} + \frac{i}{2}\bar{\chi}\gamma^\mu\partial_\mu\chi + \frac{i}{2}\bar{\psi}\gamma^\mu D_\mu\psi - \frac{1}{2}\left(m\bar{\chi}\chi + M\bar{\psi}\psi\right) + \mathcal{L}_5 + \mathcal{L}_{\geq 6}, \tag{1}$$

where $\mathcal{L}_5$ contains the dimension-5 operators

$$\mathcal{L}_5 = -\frac{1}{2}\frac{\kappa}{\Lambda}\bar{\psi}\psi H^\dagger H - \frac{1}{2}\frac{\kappa'}{\Lambda}\bar{\chi}\chi H^\dagger H - \frac{\lambda}{\Lambda}\bar{\chi}\psi^a H^\dagger\tau^a H + \text{h.c.} + \dots \tag{2}$$

Here $H$ is the SM Higgs doublet; $\psi$ is written as a column vector $(\psi_1, \psi_2, \psi_3)^T$ with $\psi^\pm = \psi_1 \pm \psi_2$ and $\psi^0 = \psi_3$. In the following, we will consider $\mathcal{L}_5$ only; dimension-6 and higher operators ($\mathcal{L}_{\geq 6}$) are neglected. Moreover, we take all parameters to be real and choose $M > 0$. Finally, since we are interested in scenarios where the DM is mostly the singlet $\chi$, we assume that $M > |m|$.

After electroweak symmetry breaking and upon replacing the Higgs field by its vacuum expectation value

$$\langle H \rangle = \begin{pmatrix} 0 \\ v \end{pmatrix}, \qquad v = 174\,\text{GeV}, \tag{3}$$

the first two terms in Eq. (2) induce a shift in the effective $\chi$ and $\psi$ mass parameters respectively. This can be absorbed through re-definitions of $m \to m + \kappa' v^2/\Lambda$ and $M \to M + \kappa v^2/\Lambda$. The third term induces a mixing between the singlet and the neutral component of the triplet. The respective mass matrix in the basis of the interaction eigenstates $(\chi, \psi_3)$ thus is

$$\mathcal{M} = \begin{pmatrix} m & -\lambda v^2/(2\Lambda) \\ -\lambda v^2/(2\Lambda) & M \end{pmatrix}. \tag{4}$$

Diagonalising this mass matrix by a unitary $2 \times 2$ matrix $R$, $\mathrm{diag}(m_{\tilde{\chi}}, m_{\tilde{\psi}^0}) = R\mathcal{M}R^\dagger$ gives mass eigenstates

$$\begin{pmatrix} \tilde{\chi} \\ \tilde{\psi}^0 \end{pmatrix} = R \begin{pmatrix} \chi \\ \psi_3 \end{pmatrix}, \quad R = \begin{pmatrix} \cos\theta & -\sin\theta \\ \sin\theta & \cos\theta \end{pmatrix}, \tag{5}$$

with physical masses

$$m_{\tilde{\chi},\tilde{\psi}^0} = \frac{1}{2}\left(m + M \mp \sqrt{(M-m)^2 + 4a^2}\right), \quad \text{where } a = \lambda v^2/(2\Lambda). \tag{6}$$

The mixing angle is given by

$$\sin 2\theta \sim 2\theta = \frac{2a}{\sqrt{(M-m)^2 + 4a^2}} \quad \rightarrow \quad \theta \approx \frac{\lambda v^2}{2\Lambda(M-m)}. \tag{7}$$

The $\psi_3 - \chi$ mixing also lifts the mass degeneracy between the charged and neutral triplet states, which would otherwise be exact at tree level. A larger effect on the $\tilde{\psi}^\pm$ mass[3] however comes from electroweak loops, increasing it by about 160 MeV [36, 37]. Since the precise $\tilde{\psi}^\pm - \tilde{\psi}^0$ mass splitting is important for phenomenology, we compute $m_{\tilde{\psi}^\pm}$ at the 2-loop level following the parametrization of [37].

**Interactions with gauge bosons:** The interaction with gauge bosons comes from the development of the covariant derivative $i\bar{\psi}\gamma^\mu D_\mu \psi \supset -g\bar{\psi}\gamma^\mu W_\mu^a T^a \psi$ in Eq. (1). After writing out the three matrix generators in the adjoint representation and developing the neutral $W$ and $\chi - \psi^3$ in their mass eigenstates, we get the following vertices:

$$\mathcal{L}_{\gamma\tilde{\psi}^+\tilde{\psi}^-} = g s_W \bar{\tilde{\psi}}^+ \gamma^\mu A_\mu \tilde{\psi}^+ - g s_W \bar{\tilde{\psi}}^- \gamma^\mu A_\mu \tilde{\psi}^-, \tag{8}$$

$$\mathcal{L}_{Z^0\tilde{\psi}^+\tilde{\psi}^-} = g c_W \bar{\tilde{\psi}}^+ \gamma^\mu Z_\mu^0 \tilde{\psi}^+ - g c_W \bar{\tilde{\psi}}^- \gamma^\mu Z_\mu^0 \tilde{\psi}^-, \tag{9}$$

$$\mathcal{L}_{W^\pm\tilde{\psi}^\mp\tilde{\chi}} = -g \sin\theta \, \bar{\tilde{\chi}} \gamma^\mu W_\mu^+ \tilde{\psi}^- + g \sin\theta \, \bar{\tilde{\chi}} \gamma^\mu W_\mu^- \tilde{\psi}^+ + \text{h.c.}, \tag{10}$$

$$\mathcal{L}_{W^\pm\tilde{\psi}^\mp\tilde{\psi}^0} = g \cos\theta \, \bar{\tilde{\psi}}^0 \gamma^\mu W_\mu^+ \tilde{\psi}^- - g \cos\theta \, \bar{\tilde{\psi}}^0 \gamma^\mu W_\mu^- \tilde{\psi}^+ + \text{h.c.}, \tag{11}$$

where $s_W = \sin\theta_W$ and $c_W = \cos\theta_W$, with $\theta_W$ being the Weinberg angle. It is worth noticing that the neutral particles do not interact with the $Z$ boson because $\psi$ is a SU(2)$_L$ triplet with zero hypercharge.

**Interactions with the Higgs boson:** These are generated after electroweak symmetry breaking through the three terms in Eq. (2). We get:

$$\mathcal{L}_{\tilde{\chi}\tilde{\chi}h} = \frac{v}{\sqrt{2}\Lambda}\left(-\frac{\lambda}{2}\sin 2\theta + \kappa \sin^2\theta + \kappa' \cos^2\theta\right)\bar{\tilde{\chi}}\tilde{\chi}h \sim -\frac{\lambda^2 v^3}{2\sqrt{2}\Lambda^2(M-m)}\bar{\tilde{\chi}}\tilde{\chi}h, \tag{12}$$

$$\mathcal{L}_{\tilde{\psi}^0\tilde{\psi}^0h} = \frac{v}{\sqrt{2}\Lambda}\left(\frac{\lambda}{2}\sin 2\theta + \kappa \cos^2\theta + \kappa' \sin^2\theta\right)\bar{\tilde{\psi}}^0\tilde{\psi}^0h \sim \frac{\lambda^2 v^3}{2\sqrt{2}\Lambda^2(M-m)}\bar{\tilde{\psi}}^0\tilde{\psi}^0h, \tag{13}$$

$$\mathcal{L}_{\tilde{\chi}\tilde{\psi}^0h} = \frac{v}{\sqrt{2}\Lambda}\left(\lambda \cos 2\theta - \kappa \sin 2\theta + \kappa' \sin 2\theta\right)\bar{\tilde{\chi}}\tilde{\psi}^0h \sim \frac{\lambda v}{\sqrt{2}\Lambda}\bar{\tilde{\chi}}\tilde{\psi}^0h, \tag{14}$$

$$\mathcal{L}_{\tilde{\psi}^+\tilde{\psi}^-h} = \frac{2\kappa v}{\sqrt{2}\Lambda}\bar{\tilde{\psi}}^-\tilde{\psi}^-h. \tag{15}$$

---

[3]Here and in the following, all odd-sector physical particles are denoted with a tilde.

**Decays into pions:**  Including electroweak corrections, the mass hierarchy in our model is $m_{\tilde{\psi}^{\pm}} > m_{\tilde{\psi}^0} > m_{\tilde{\chi}}$. The $\tilde{\psi}^{\pm}$ can thus decay either to the $\tilde{\chi}$ or to the $\tilde{\psi}^0$. Both transitions proceed via a virtual $W$-boson, the width of $\tilde{\psi}^{\pm} \to \tilde{\chi}(W^{\pm})^*$ being suppressed by the small mixing, and the width of $\tilde{\psi}^{\pm} \to \tilde{\psi}^0(W^{\pm})^*$ being suppressed by the tiny mass splitting. For the parameter ranges of interest for this study, the $\tilde{\psi}^{\pm}$ is thus often long-lived at collider scales.

Given that the $\tilde{\psi}^{\pm}$–$\tilde{\psi}^0$ mass difference is only $O(160)$ MeV, hadronic $\tilde{\psi}^{\pm} \to \tilde{\psi}^0(W^{\pm})^*$ transitions have to be treated as decays into pions, $\tilde{\psi}^{\pm} \to \tilde{\psi}^0 \pi^{\pm}$, instead of decays into quarks, $\tilde{\psi}^{\pm} \to \tilde{\psi}^0 qq'$ [38]. We implement this via a non-perturbative $W$–$\pi$ mixing [39]

$$\mathcal{L}_{W\pi} = \frac{g f_\pi}{2\sqrt{2}} W_\mu^+ \partial^\mu \pi^- + \text{h.c.}, \tag{16}$$

where $f_\pi = 130$ MeV is the pion decay constant. This gives an effective $\tilde{\psi}^{\pm}\tilde{\psi}^0\pi^{\mp}$ interaction of the form

$$\mathcal{L}_{\tilde{\psi}^+\tilde{\psi}^0\pi^-} = \frac{g^2 \cos\theta f_\pi}{2\sqrt{2}m_W^2} \bar{\tilde{\psi}}^0 \gamma^\mu \partial_\mu \pi^- \tilde{\psi}^+, \tag{17}$$

which we use to compute the 2-body decay width $\Gamma(\tilde{\psi}^{\pm} \to \tilde{\psi}^0 \pi^{\pm})$ in CalcHEP. Indeed, $\tilde{\psi}^{\pm} \to \tilde{\psi}^0 \pi^{\pm}$ is often the dominant decay mode and determines the lifetime of the $\tilde{\psi}^{\pm}$. This will be relevant later for the LHC constraints on the model.

# 3   Relic density calculation

Since for small couplings the particles of the dark sector might not be in thermal equilibrium with each other, separate equations for the evolution of their abundances must be written. In the case of the singlet-triplet model considered in this paper, we define two dark sectors, sector 1 containing the singlet $\tilde{\chi}$ and sector 2 containing the triplet $\tilde{\psi}^{\pm}, \tilde{\psi}^0$ states. In addition, SM particles are assigned to sector 0.

We will always take the singlet as the lightest dark particle and thus the DM candidate. The lightest component of the triplet, which is also odd under $\mathbb{Z}_2$, will decay to the DM and SM particles. The charged component of the triplet has electromagnetic interactions and is therefore in thermal equilibrium with the SM. Moreover, processes such as $\tilde{\psi}^{\pm} \text{SM} \leftrightarrow \tilde{\psi}^0 \text{SM}$ are always efficient so that all particles of sector 2 are in thermal equilibrium in the early Universe; they thus have the same abundance $Y_2$. The couplings of the singlet can be much weaker and the evolution of its abundance, $Y_1$, must be solved independently. The general equations for the evolution of the abundances with temperature $T$ read

$$
\begin{aligned}
\frac{dY_1}{dT} &= \frac{1}{3H}\frac{ds}{dT}\Bigg[ \langle\sigma_{1100}v\rangle(Y_1^2 - Y_1^{eq2}) + \langle\sigma_{1122}v\rangle\left(Y_1^2 - Y_2^2\frac{Y_1^{eq2}}{Y_2^{eq2}}\right) \\
&\quad + \langle\sigma_{1200}v\rangle(Y_1 Y_2 - Y_1^{eq}Y_2^{eq}) + \langle\sigma_{1222}v\rangle\left(Y_1 Y_2 - Y_2^2\frac{Y_1^{eq}}{Y_2^{eq}}\right) \\
&\quad - \langle\sigma_{1211}v\rangle\left(Y_1 Y_2 - Y_1^2\frac{Y_2^{eq}}{Y_1^{eq}}\right) - \frac{\Gamma_{2\to1}}{s}\left(Y_2 - Y_1\frac{Y_2^{eq}}{Y_1^{eq}}\right)\Bigg],
\end{aligned}
\tag{18}
$$

$$\frac{dY_2}{dT} = \frac{1}{3H}\frac{ds}{dT}\left[\langle\sigma_{2200}v\rangle(Y_2^2 - Y_2^{eq2}) - \langle\sigma_{1122}v\rangle\left(Y_1^2 - Y_2^2\frac{Y_1^{eq2}}{Y_2^{eq2}}\right)\right.$$

$$+ \langle\sigma_{1200}v\rangle(Y_1 Y_2 - Y_1^{eq}Y_2^{eq}) - \langle\sigma_{1222}v\rangle\left(Y_1 Y_2 - Y_2^2\frac{Y_1^{eq}}{Y_2^{eq}}\right)$$

$$\left. + \langle\sigma_{1211}v\rangle\left(Y_1 Y_2 - Y_1^2\frac{Y_2^{eq}}{Y_1^{eq}}\right) + \frac{\Gamma_{2\to1}}{s}\left(Y_2 - Y_1\frac{Y_2^{eq}}{Y_1^{eq}}\right)\right], \tag{19}$$

where $Y_i^{eq}$ are the equilibrium abundances, $H$ is the Hubble parameter, $\langle\sigma_{\alpha\beta\gamma\delta}v\rangle$ are the thermally averaged cross-sections for processes involving annihilation of particles of sectors $\alpha\beta \to \gamma\delta$. In general, the thermally averaged cross-section is given by

$$\langle\sigma_{\alpha\beta\gamma\delta}v\rangle = \frac{1}{C_{\alpha\beta}\bar{n}_\alpha\bar{n}_\beta}\sum_{abcd}\frac{Tg_a g_b}{8\pi^4}\int\sqrt{s}p_{ab}^2(s)K_1(\frac{\sqrt{s}}{T})C_{ab}\sigma_{ab\to cd}(s)ds, \tag{20}$$

where $C_{ab} = 1/2$ if $a = b$ and 1 otherwise; the sum runs over all particles in a given sector, $a \in \alpha, b \in \beta, c \in \gamma, d \in \delta$, when $\alpha = \beta$ (or $\gamma = \delta$) then the additional condition applies $a \leq b$ (or $c \leq d$). $\bar{n}_\alpha$ is the equilibrium number density which for non-relativistic particles reads

$$\bar{n}_\alpha = s(T)Y_\alpha^{eq} = \frac{T}{2\pi^2}\sum_{a\in\alpha}g_a m_a^2 K_2(\frac{m_a}{T}). \tag{21}$$

The entropy $s$ given by

$$s = \frac{2\pi^2}{45}h_{\text{eff}}T^3, \tag{22}$$

with $h_{\text{eff}}$ the effective number of degrees of freedom. Note that $Y_0^{eq} = 0.238$ for the SM sector.

The conversion term $\Gamma_{2\to1}$ in Eqs. (18) and (19) includes both the co-scattering term as well as a decay term:

$$\Gamma_{2\to1} = \frac{\sum_{a\in2}\Gamma_{a\to1,0}\,g_a m_a^2 K_1\left(\frac{m_a}{T}\right) + \sum_{a\in1}\Gamma_{a\to2,0}\,g_a m_a^2 K_1\left(\frac{m_a}{T}\right)}{\sum_{a\in2}g_a m_a^2 K_2\left(\frac{m_a}{T}\right)} + \langle\sigma_{2010}v\rangle\bar{n}_0, \tag{23}$$

where $\Gamma_{a\to1,0}$ is the decay width of particle $a$ of sector 2 into particles of sectors 1 and 0. The processes included in the width calculation correspond to the decays into one particle of sector 1 and up to 3 particles of sector 0. $\Gamma_{a\to2,0}$ is defined analogously. However, the second term in the numerator of Eq. (23) does not exist in our model which contains only one stable particle in sector 1. All these terms are included in the function darkOmegaN of MICROMEGAS described in Appendix A. Note that by default, when 2-body decays are present, the 3-body processes are not computed by MICROMEGAS. However, there is a switch to include 3-body processes in all cases as explained in the appendix. This is important in our model, as 3-body decays of $\tilde{\psi}^\pm \to \tilde{\chi}\bar{f}f'$ compete with the 2-body decay $\tilde{\psi}^\pm \to \tilde{\psi}^0\pi^\pm$.

The total relic density is obtained after solving Eqs. (18) and (19) for the abundances today:

$$\Omega h^2 = 2.742 \times 10^8\left(m_{\tilde{\chi}}Y_1 + m_{\tilde{\psi}^0}Y_2\right). \tag{24}$$

In the STFM, the triplet states decay fast enough such that the only relevant contribution to the relic density is from the term $Y_1$ in Eq. (24).

When the couplings are large enough such that both sectors are in thermal equilibrium, then $Y_1/Y_2 = Y_1^{eq}/Y_2^{eq}$ and Eqs. (18), (19) simplify considerably to recover the usual freeze-out

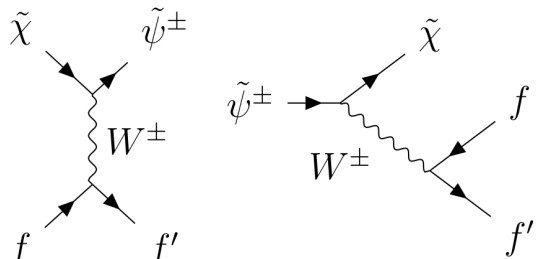

Figure 1: Dominant processes contributing to the co-scattering (left) and decay (right) terms in the STFM.

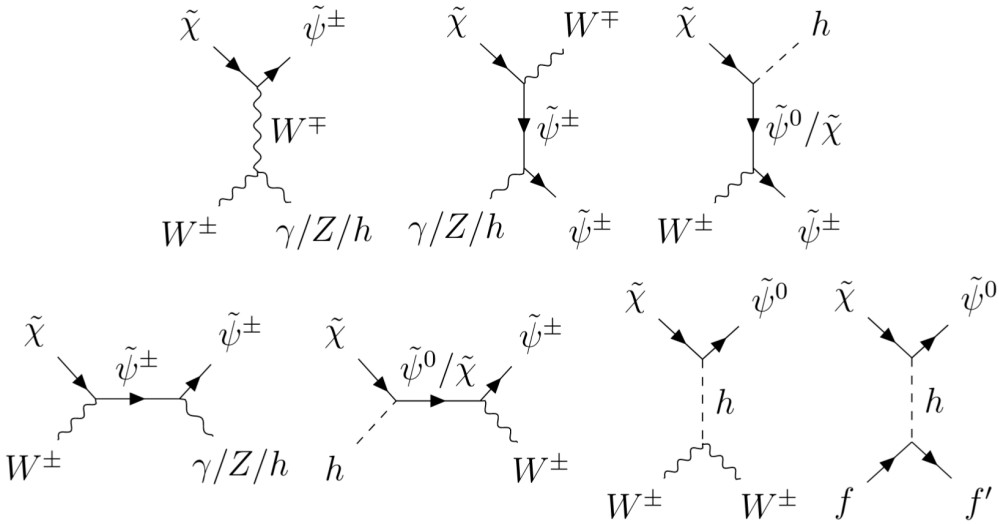

Figure 2: Subdominant co-scattering processes in the STFM.

equations. In this case the only contributions are from $\langle\sigma_{1100}v\rangle$ for DM annihilation, as well as $\langle\sigma_{1200}v\rangle$ and $\langle\sigma_{2200}v\rangle$ which are relevant for co-annihilation processes such as $\tilde{\chi}\tilde{\psi}^0 \to W^+W^-$ or $\tilde{\psi}^0\tilde{\psi}^0 \to W^+W^-$. Solving the two abundance equations will lead to the same result as solving a single abundance equation that is $Y_2 = 0$ and $Y_1 = Y$ of the single equation. On the other hand, when the coupling of the singlet, set by the Wilson coefficient $\lambda$, is small, self-annihilation of the singlet becomes negligible and the abundance equations simplify to

$$\frac{dY_1}{dT} = \frac{-\Gamma_{2\to1}}{HT}\left[Y_2 - Y_1\frac{Y_2^{eq}}{Y_1^{eq}}\right], \tag{25}$$

$$\frac{dY_2}{dT} = \frac{s}{HT}\left[\langle\sigma_{2200}v\rangle(Y_2^2 - Y_2^{eq2}) + \frac{\Gamma_{2\to1}}{s}\left(Y_2 - Y_1\frac{Y_2^{eq}}{Y_1^{eq}}\right)\right]. \tag{26}$$

The dominant processes contributing to the co-scattering and decay terms entering $\Gamma_{2\to1}$ in the STFM are scattering on SM fermions through the exchange of a $W$-boson and $\tilde{\psi}^\pm \to \tilde{\chi} f f'$ decays via an off-shell $W$ boson ($\tilde{\psi}^\pm \to \tilde{\psi}^0\pi^\pm$ decays do not contribute to the conversion term). The relevant diagrams are shown in Fig. 1. Numerically it turns out that the decays contribute at most at the level of a few percent to obtaining $\Omega h^2 \simeq 0.12$. Nonetheless they may have a relevant effect for maintaining DM in thermal equilibrium during the evolution of the number density. Scattering on SM bosons, shown in Fig. 2, gives a subdominant contribution, also of the level of a few percent.

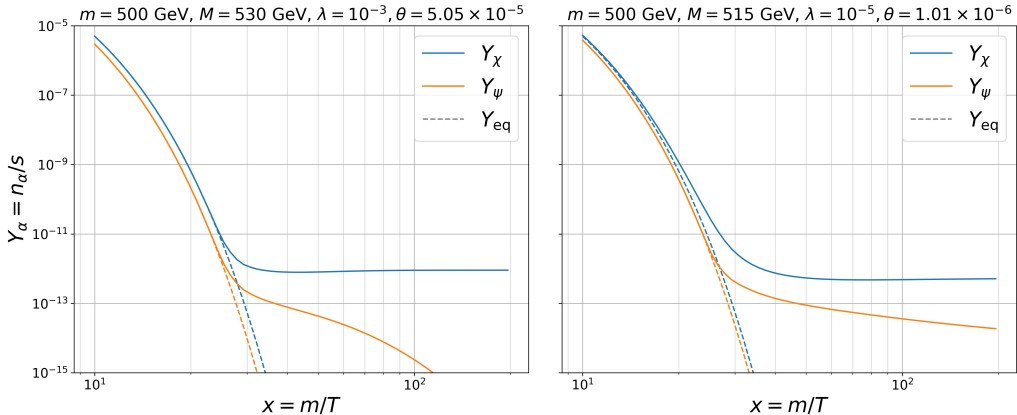

Figure 3: Evolution of the $\chi$ and $\psi$ abundances for $\lambda = 10^{-3}$ (left) and $\lambda = 10^{-5}$ (right); The $\chi$ mass parameter is set to $m = 500$ GeV, while $M$ is adjusted to obtain $\Omega h^2 \simeq 0.12$.

For illustration, we show in Fig. 3 the evolution of $Y_{\chi,\psi} = Y_{1,2}$ in the STFM for two values of $\lambda = 10^{-3}$ and $10^{-5}$, for $m = 500$ GeV and $M$ chosen such that $\Omega h^2 \simeq 0.12$. For $\lambda = 10^{-3}$, co-annihilation dominates. In this case, both sectors follow their equilibrium distribution until $x \approx 25$, where freeze-out occurs. After freeze-out, the $\psi$ rapidly decay to the DM. However, since $Y_\psi \ll Y_\chi$, the decay term gives only a small contribution to the relic density. For $\lambda = 10^{-5}$ and a smaller mass splitting, co-scattering dominates and $Y_\chi$ departs from equilibrium much sooner. In this case, the decay of $\tilde{\psi}^0$ continues until small temperatures (here, the $\tilde{\psi}^\pm$ primarily decays into $\tilde{\psi}^0$).

The early departure of $Y_\chi$ from the equilibrium distribution in the right panel of Fig. 3 occurs because chemical equilibrium between $\chi$ and $\psi$, maintained by co-scattering and decay processes at high enough $T$, is lost. Quantitatively this happens when $\Gamma_{2\to1}$ is not much larger than the Hubble rate. To illustrate this in more detail, we compute $\Gamma_{2\to1}/H(T)$ for the decay and co-scattering contributions separately. The result is shown in Fig. 4 as a function of $x = m/T$, on the left for $\lambda = 10^{-3}$ and on the right for $\lambda = 10^{-5}$. As can be seen, for $\lambda = 10^{-3}$, both types of processes maintain equilibrium ($\Gamma/H \gg 1$) until after the freeze-out of $\psi$. However, for $\lambda = 10^{-5}$, $\Gamma_{2\to1}/H(T)$ is $\mathcal{O}(1)$ at freeze-out, thus a treatment using the one-component Boltzmann equation is not appropriate. We also see that co-scattering is more efficient at higher temperatures and decreases with increasing $x$. For completeness and as a reference, Fig. 4 also shows $\Gamma/H(T)$ for $\psi\psi \to$ SM SM annihilation.

Inelastic scattering processes $\chi$ SM $\leftrightarrow \psi$ SM and (inverse) decays $\psi \leftrightarrow \chi$ SM are also involved in maintaining kinetic equilibrium. As already mentioned in the Introduction, deviations from kinetic equilibrium are expected when the DM couplings become too weak [34]. In this case, the DM distribution at freeze-out does not follow a Maxwell-Boltzmann distribution and a complete treatment involves solving the full momentum-dependent Boltzmann equation. Deviations from kinetic equilibrium in the co-scattering region were shown to have a mild (∼10%) impact on the final relic density in the scenario considered in [10] (Appendix C). In [30] it was argued that there can be larger effects in the STFM because decay processes are less important than in the model considered in [10]; however, this study also made some simplifying assumptions, e.g., ignoring decays and the subdominant co-scattering processes. To reliably quantify the uncertainty introduced by using the integrated Boltzmann equations, a one-to-one comparison with the full, unintegrated approach would be needed. A complete solution to the unintegrated Boltzmann equation within MICROMEGAS is however beyond the scope of this work.

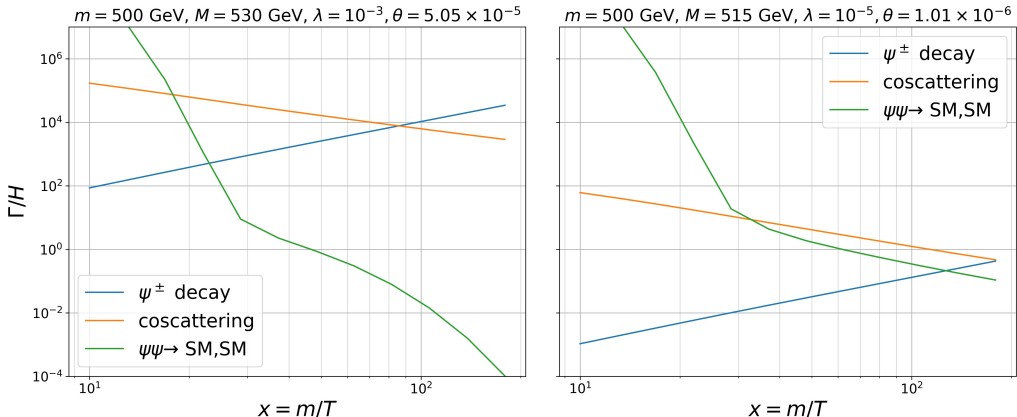

Figure 4: Contributions to the DM equilibrium for $\lambda = 10^{-3}$ (left) and $\lambda = 10^{-5}$ (right). Different contributions are all converted into reaction rates $\Gamma_i$ and compared to the universe expansion rate $H$; the $\chi$ mass parameter is set to $m = 500$ GeV, while $M$ is adjusted to obtain $\Omega h^2 \simeq 0.12$.

## 4    Numerical analysis

Let us now turn to the numerical analysis of the STFM parameter space. To this end, we take the mass parameters $m$ and $M$ together with the Wilson coefficient $\lambda$ from Eqs. (1) and (2) as input parameters, fixing $\kappa = \kappa' = 0$ and $\Lambda = 10$ TeV. For small $\lambda$, as relevant in this study, the singlet-triplet mixing is small and $m \simeq m_{\tilde{\chi}}$. For each choice of $(m, \lambda)$, we scan over $M$ to obtain $\Omega h^2 = 0.12$ [6]. This fixes the $\tilde{\psi}^0$–$\tilde{\chi}$ mass difference $\Delta m \equiv m_{\tilde{\psi}^0} - m_{\tilde{\chi}}$. The mixing angle is then $\theta \approx \lambda \times 1.5\,\text{GeV}/\Delta m$, cf. Eq. (7).

The $\tilde{\psi}^\pm$ mass is given by $m_{\tilde{\psi}^\pm} = M + \delta m_\psi^{\text{2loop}}$, where $\delta m_\psi^{\text{2loop}}$ are electroweak corrections at the 2-loop level as parametrized in [37]; they lead to a small mass splitting between the $\tilde{\psi}^\pm$ and the $\tilde{\psi}^0$ of about $150 - 165$ MeV depending on $M$. The precise value of this mass splitting is crucial for the mean lifetime $c\tau_0(\tilde{\psi}^\pm)$, which in turn determines the LHC signatures.

To evaluate LHC constraints, we use SMODELS v2.2.0 [40–44], interfaced to MICROMEGAS [45, 46]. This interface automatically creates the input file for SMODELS including all relevant LHC production cross sections at $\sqrt{s} = 8$ and 13 TeV (computed with CALCHEP) and writes out the most constraining result. The collider signatures of the STFM resemble those of chargino/neutralino production in the MSSM with small mass splitting between the wino and bino states. Relevant LHC constraints therefore come from searches for long-lived charginos, in particular from disappearing track searches, for which SMODELS has the ATLAS and CMS searches [47, 48] from Run 2 implemented. Searches for promptly decaying charginos/neutralinos do not give any relevant constraints for the small mass splittings relevant here. Searches for heavy stable charged particles are also not effective, because the $\tilde{\psi}^\pm$ mean decay length does not exceed $\mathcal{O}(10)$ cm in the parameter range we consider.

To illustrate the importance of co-scattering for obtaining the correct relic density, $\Omega h^2 = 0.12$, we introduce two quantities, $\Delta_{1s}^\Omega$ and $\Delta_{2s}^\Omega$. The former is the fractional difference of the relic densities obtained in the 1-sector or 2-sector computations:

$$\Delta_{1s}^\Omega \equiv 1 - \frac{\Omega h^2(1\,\text{sector})}{\Omega h^2(2\,\text{sectors})}. \tag{27}$$

More explicitly, $\Omega h^2(1\,\text{sector})$ is the value obtained when using the standard `darkOmega` function of MICROMEGAS, which involves only one Boltzmann equation and thus includes only co-annihilation. In contrast, $\Omega h^2(2\,\text{sectors})$ is the value obtained by means of `darkOmegaN` with

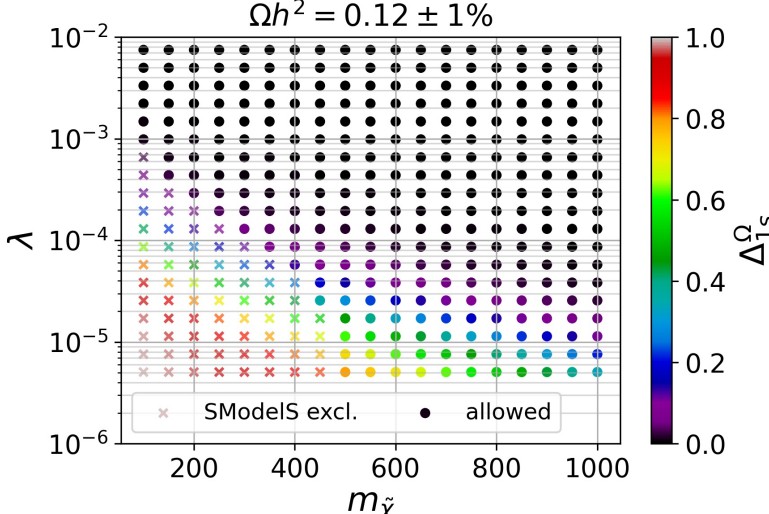

Figure 5: Scan result in the plane of $\lambda$ versus DM mass $m_{\tilde{\chi}}$. For each point, the triplet mass parameter $M$ is adjusted such that $\Omega h^2 = 0.12$. The colour scale indicates $\Delta^{\Omega}_{1s}$ as defined in Eq. (27). Full-coloured points pass collider constraints, crosses are excluded by the disappearing track results in SMODELS v2.2.0.

the dark particles split in two sectors, i.e. from solving the coupled system of two Boltzmann equations including all co-scattering and decays processes. A value of $\Delta^{\Omega}_{1s} = 0.5$ means that the conventional calculation with one Boltzmann equation gives a result which is a factor of 2 too small, and one would conclude that the DM candidate is under-abundant in the scenario at hand. The second quantity, $\Delta^{\Omega}_{2s}$, is defined as

$$\Delta^{\Omega}_{2s} \equiv 1 - \frac{\Omega h^2 (2 \text{ sectors})}{\Omega h^2 (2 \text{ sectors, no co-scattering})} \,. \tag{28}$$

Here $\Omega h^2 (2 \text{ sectors})$ is the relic density from solving two Boltzmann equations as above, while $\Omega h^2 (2 \text{ sectors, no co-scattering})$ is the value in the same approach when the co-scattering processes (but not the decays) are neglected. The latter can be computed in MICROMEGAS via the `ExcludedFor2DM="2010"` command as explained in Appendix A. Note that $\Delta^{\Omega}_{2s} = 0.5$ means that $\Omega h^2$ increases by a factor 2 when co-scattering processes are neglected, while $\Delta^{\Omega}_{2s} = 0.9$ means an increase by a factor 10.

Figures 5 and 6 show the scan results in the plane of $\lambda$ versus $m_{\tilde{\chi}}$. At each point in the plots, $\Delta m \equiv m_{\tilde{\psi}^0} - m_{\tilde{\chi}}$ is adjusted such that $\Omega h^2 = 0.12$ within 1% precision. While the full-coloured points pass collider constraints, the points marked as crosses are excluded by the disappearing track results in SMODELS. In order to focus on the transition from the co-annihilation to the co-scattering regimes, we consider the range $\lambda = [10^{-2}, 5 \times 10^{-6}]$. For smaller values of $\lambda$ the equilibrium condition becomes questionable and the calculation in MICROMEGAS may not be valid any more (see the discussion at the end of section 3). Moreover, the $\tilde{\psi}^0$ becomes very long lived, such that constraints from Big Bang Nucleosynthesis (BBN) may become relevant. In fact, for small DM masses around 100 GeV, $c\tau_0(\tilde{\psi}^0) > 100$ sec at leading order even for $\lambda \lesssim 10^{-5}$. However, this region is excluded by LHC constraints, so we do not consider BBN bounds in our analysis.

In Fig. 5, the colour code shows $\Delta^{\Omega}_{1s}$ as defined in Eq. (27). This indicates the importance of solving two Boltzmann equations instead of just one. We see that, for $m_{\tilde{\chi}}$ around 100–200 GeV, the splitting into two dark sectors ($1 = \tilde{\chi}$ and $2 = \tilde{\psi}^{\pm}, \tilde{\psi}^0$) is relevant already at $\lambda \sim (\text{a few}) \times 10^{-4}$. With increasing mass, the importance of the two-dark-sectors

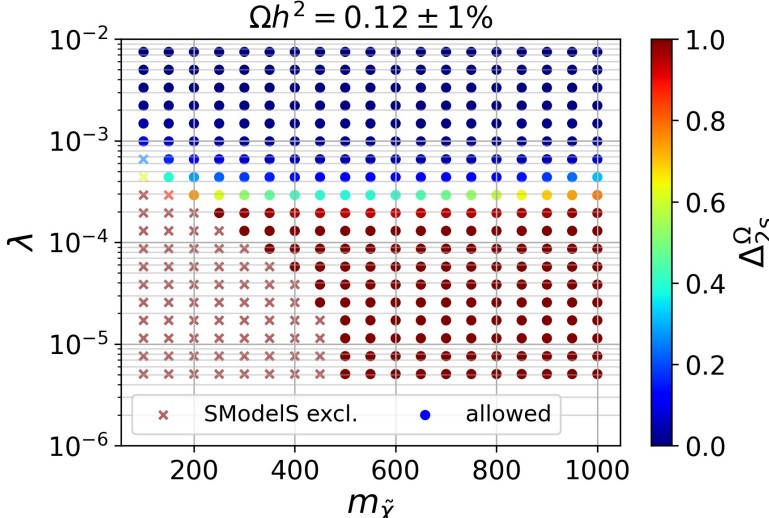

Figure 6: Same as Fig. 5 but with the colour scale showing $\Delta_{2s}^{\Omega}$ as defined in Eq. (28).

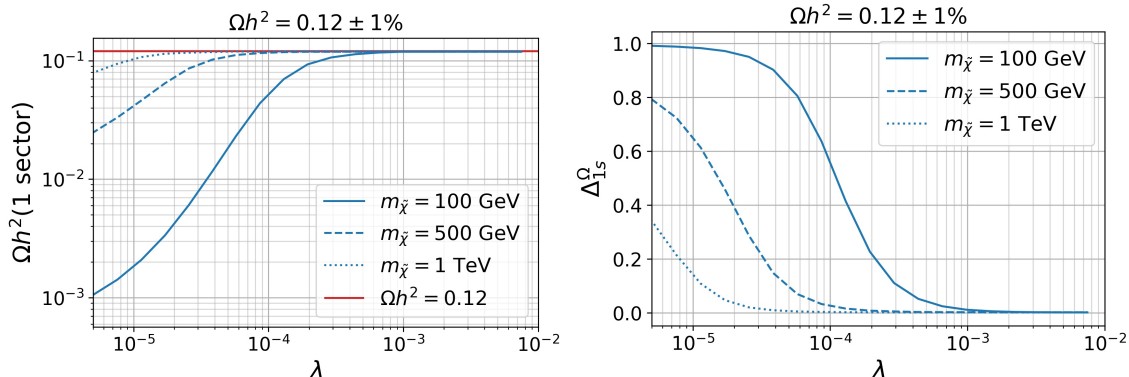

Figure 7: On the left $\Omega h^2$(1 sector), on the right $\Delta_{1s}^{\Omega}$ as function of $\lambda$ for $m_{\tilde{\chi}} = 100$, 500 and 1000 GeV; at each point, the triplet mass parameter $M$ is adjusted such that $\Omega h^2 = 0.12$ in the full 2-sectors calculation.

treatment sets in at smaller $\lambda$. However even at $m_{\tilde{\chi}} = 1$ TeV, there is a large effect for $\lambda \lesssim 10^{-5}$. Roughly, the black points correspond to the co-annihilation dominated region, while the colourful points correspond to the co-scattering domain. The current LHC constraints challenge the co-scattering region for DM masses up to about 450 GeV, actually excluding most of this region. In the co-annihilation region, LHC constraints are not effective.

In Fig. 6, the colour code shows $\Delta_{2s}^{\Omega}$ as defined in Eq. (28). This illustrates the importance of the co-scattering term in the two-dark-sectors treatment. We observe that for $\lambda$ of the order of $10^{-2}$–$10^{-3}$ (dark blue points), the decay processes are sufficient to keep the two dark sectors in equilibrium. This rapidly changes with decreasing $\lambda$, and from $\lambda \approx 3 \times 10^{-4}$ onwards the final relic density is dominated by the co-scattering processes. This conclusion depends very little on $m_{\tilde{\chi}}$.

The behaviour of $\Omega h^2$(1 sector) and $\Delta_{1s}^{\Omega}$ as function of $\lambda$ is shown explicitly in Fig. 7 for three choices of DM mass, $m_{\tilde{\chi}} = 100$, 500 and 1000 GeV. Analogously, Fig. 8 shows the behaviour of $\Omega h^2$(2 sectors, no co-scattering) and $\Delta_{2s}^{\Omega}$ for the same choice of masses.

The singlet-triplet mass difference needed to achieve $\Omega h^2 = 0.12$ is shown in Fig. 9 as a function of $m_{\tilde{\chi}}$, for the same range of $\lambda$ as in Figs. 5 and 6. When co-annihilation is dominant

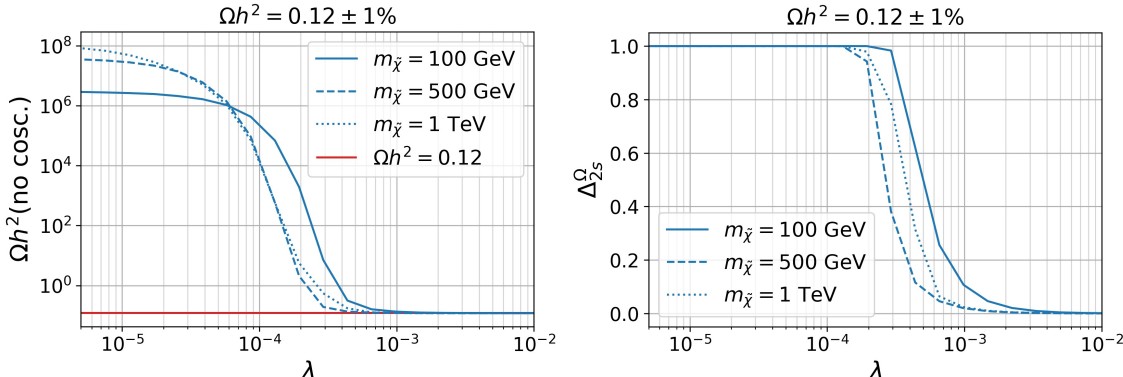

Figure 8: As Fig. 7 but showing on the left $\Omega h^2$(2 sectors, no co-scattering) and on the right the corresponding $\Delta_{2s}^{\Omega}$.

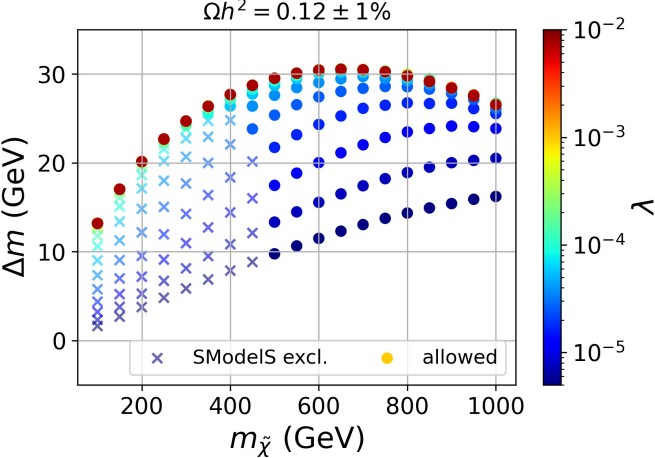

Figure 9: Illustration of mass difference $\Delta m$ needed to obtain $\Omega h^2 = 0.12$. Points marked as crosses are excluded by SMODELS v2.2.0.

(dark red points), a finely adjusted mass difference in the range of about 10–30 GeV is needed. The relative mass difference $\Delta m/m_{\tilde{\chi}}$ steadily decreases from 13% at $m_{\tilde{\chi}} = 100$ GeV to about 3% at $m_{\tilde{\chi}} = 1$ TeV. This well-known behaviour does not depend on $\lambda$ as long as co-annihilation is dominant. In the co-scattering phase, however, smaller couplings require smaller mass differences in order for $\tilde{\chi}$ SM $\to \tilde{\psi}$ SM processes to remain efficient. This opens a new region of smaller mass splittings in the parameter space of the model, where the cosmologically observed DM abundance can be saturated. Without the co-scattering mechanism, one would conclude that the relic density in this region was too small and $\tilde{\chi}$ could constitute only part of the DM. As before, we also see that long-lived particle searches at the LHC exclude a large part of the co-scattering region for DM masses up to about 450 GeV.

The interdependence of $\Delta m$, the DM coupling and the importance of co-annihilation or co-scattering is further illustrated in Fig. 10 (top panels). This figure presents the scan points in the plane of mixing angle $\theta$ vs. singlet-triplet mass difference $\Delta m$, for $m_{\tilde{\chi}} = 100$–600 GeV. The left-most line of points is for $m_{\tilde{\chi}} = 100$ GeV, the right-most is for $m_{\tilde{\chi}} = 600$ GeV; in-between $m_{\tilde{\chi}}$ increases in steps of 50 GeV. We see again that, as long as co-annihilation is dominant, for a given $m_{\tilde{\chi}}$, $\Delta m$ is almost constant as the mixing decreases. Once co-scattering takes over, smaller mixing also means smaller $\Delta m$ to achieve the correct relic density. $\theta$ saturates just below $10^{-6}$, as it is inversely proportional to $\Delta m$, see Eq. (7). The small mixing and small

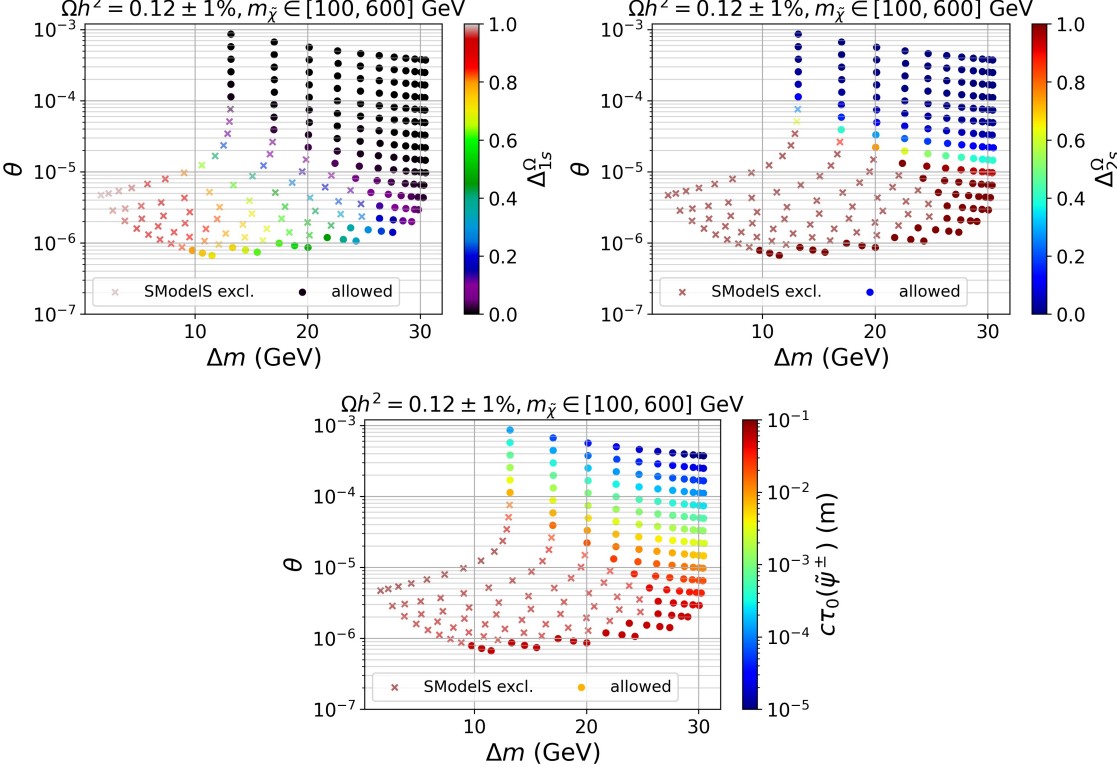

Figure 10: Mixing angle $\theta$ versus $\Delta m$ for $m_\chi = 100 - 600$ GeV (from left to right in steps of 50 GeV). The colour scales show $\Delta_{1s}^\Omega$ (top left), $\Delta_{2s}^\Omega$ (top right), and the $\tilde\psi^\pm$ mean decay length, $c\tau_0(\tilde\psi^\pm)$, in meter (bottom). Points marked as crosses are excluded by SMODELS v2.2.0.

mass differences make the $\tilde\psi^\pm$ long-lived in much of the co-scattering region, as can be seen in the bottom panel of Fig. 10. The associated decay lengths are of the order of 1–10 cm. It is this behaviour that causes constraints from disappearing track searches at the LHC to kick in.

Before concluding this analysis, we note that the computed $\Omega h^2$ is subject to systematic uncertainties, stemming in part from the usage of the momentum-integrated Boltzmann equations.[4] Quantifying this uncertainty would be a full-fledged study in itself, beyond the scope of this work. Instead, we show in Fig. 11 the effect of an assumed 10% theoretical uncertainty. As can be seen, this results in a widening of the range of $\Delta m$, but does not qualitatively change our results. In particular the turn-over to smaller $\Delta m$ that indicates the transition from the co-annihilation to the co-scattering regime in the left panel of Fig. 11 is hardly affected. We also note that the bands of $\Omega h^2 = 0.12 \pm 10\%$ are narrower in the co-scattering region than in the co-annihilation region.

# 5 Conclusions

In scenarios with very small DM couplings and small mass splittings between the DM and other dark-sector particles, so-called "co-scattering" or "conversion-driven freeze-out" can be the dominant mechanism for DM production. Characteristic for this mechanism is that freeze-out takes place out of chemical equilibrium. Self-annihilation of DM is too inefficient to achieve $\Omega h^2 \approx 0.1$. Instead, inelastic scattering processes of the type $\chi\, \mathrm{SM} \leftrightarrow \psi\, \mathrm{SM}$ are primarily

---

[4]Higher-order loop corrections will also be relevant.

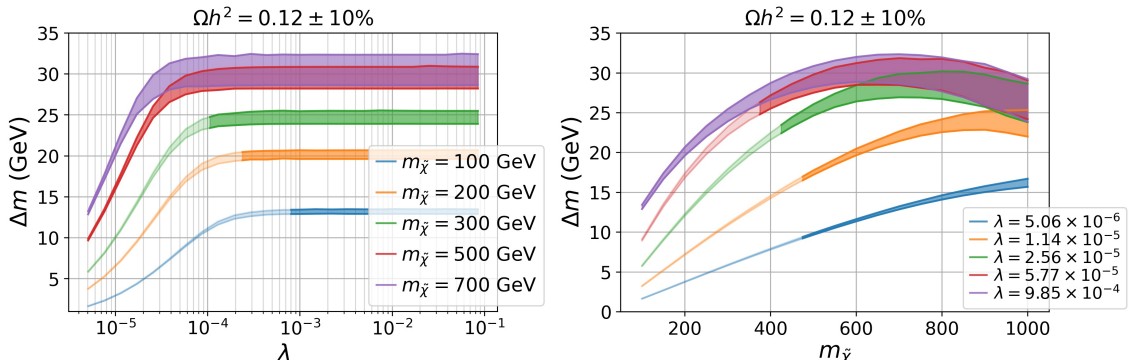

Figure 11: Bands of $\Omega h^2 = 0.12 \pm 10\%$ in the plane of $\Delta m$ vs $\lambda$ for several DM masses (left) and in the plane of $\Delta m$ vs $m_{\tilde{\chi}}$ for several values of $\lambda$ (right); the light-shaded areas are excluded by SMODELS v2.2.0.

responsible for DM formation. Moreover, decays and inverse decays of dark sector particles, like $\psi \leftrightarrow \chi$ SM, also have to be taken into account in the conversion terms of the Boltzmann equations.

We presented the first inclusion of this mechanism in a public DM tool, MICROMEGAS. The numerical treatment relies heavily on the machinery for multi-component DM in MICROMEGAS [32, 33] as it requires to solve at least two separate Boltzmann equations, one for each set of dark particles in thermal equilibrium. To illustrate both, the new capabilities of MICROMEGAS as well as the phenomenological implications of the co-scattering regime, we performed a case study for the singlet-triplet fermion model, STFM. This model extends the SM by two electroweak multiplets, a singlet $\chi$ and a triplet $\psi$, which are both odd under a new $\mathbb{Z}_2$ symmetry, while the SM particles are even. The $\chi$-like state is the DM candidate; it has very weak couplings induced by a small mixing with triplet. Our numerical analysis concentrated on the transition between the co-annihilation and the co-scattering regimes, and we showed that the latter can open up a new region in the parameter space of the model.

The charged triplet states, $\tilde{\psi}^\pm$, are typically long-lived in the co-scattering region, leading to distinct collider signatures. Using SMODELS v2.2.0 to evaluate the current LHC constraints, we found that disappearing track searches exclude DM masses up to $m_{\tilde{\chi}} \approx 200$ GeV in the transition region between co-annihilation and co-scattering, and up to $m_{\tilde{\chi}} \approx 450$ GeV for very small $\lambda$, where co-scattering dominates. A precise calculation of the mass splitting among the triplet-like states as well as the inclusion of $\tilde{\psi}^\pm \to \tilde{\chi} \pi^\pm$ decays are important to that end.

The new version of MICROMEGAS, v5.3.35, is publicly available at https://lapth.cnrs.fr/micromegas/. It includes the STFM implementation together with a README explaining its usage, and a demo program (demo.c) illustrating some of the new functionalities presented in this paper. It also includes an updated interface to SMODELS v2.2 [46]. The new MICROMEGAS routines relevant for co-scattering are described in detail in the appendix.

An important caveat is that, so far, MICROMEGAS employs momentum-integrated Boltzmann equations only. For a precise calculation of the DM relic density including the effects of early kinetic decoupling, the full momentum-dependent Boltzmann equations should be solved, as advertised in [30]. This is left for future work.

# Acknowledgments

We thank Felix Brümmer for discussions and comparisons in the early stage of this work. G.A. moreover acknowledges discussions with Jan Heisig and the hospitality of the RWTH Aachen

during a research visit related to this work.

This work was supported in part by the CNRS and RFBF, project number 20-52-1500, and by the IN2P3 master project "Théorie – BSMGA".

# A  MicrOMEGAs routines for co-scattering

For the computation of co-scattering, as in the case of $N$-component DM, the dark particles need to be divided into *sectors*, within each of which chemical equilibrium is observed. By default, this separation is defined by the number of ~ symbols in the beginning of the particle names. Thus, ~x1 and ~~x1 denote dark particles of two different sectors. Usually, the sector assignment corresponds to the charge of the discrete symmetry responsible for DM stability, cf. section 2 in the MICROMEGAS manual. However, in the absence of chemical equilibrium, the splitting into sectors needs to be done differently. To this end, the function

• defThermalSet(n,particles_list) moves all particles mentioned in *particles_list* to sector $n$. All particles that were assigned to sector $n$ before this command are returned to their default sectors specified by the number of ~ in the beginning of their names. Particles in the *particles_list* have to be separated by commas, and particle and anti-particle automatically belong to the same sector. By definition, sector 0 is the SM bath while sector $-1$ is used to define *feeble* particles which do not take part in freeze out. Such particles will be ignored when solving for the relic density. Sectors $n > 0$ are used for all other cases.

In general, defThermalSet can define a set which includes particles with different charges of the discrete symmetry group (different number of ~ symbols) — in particular the set could include $\mathbb{Z}_2$odd particles as well as SM particles. In this case the user must keep in mind that the abundance equations are solved for sectors $n > 0$ only. This entails that a $Z_2$ odd particle assigned to sector 0 will not be considered as potential DM candidate. The function returns an error code if *particles_list* contains a particle name which is not defined in the model.

• printThermalSets() prints the contents of all particle sets specified by defThermalSet on the screen.

To verify whether chemical equilibrium is reached in one sector, one can use

• checkTE(n,T,mode,Beps) which checks the condition for chemical equilibrium in the $n^{th}$ sector at temperature T. If mode=0, then both decay and co-scattering are taken into account. If mode=1 (2), then only decay (co-scattering) processes are taken into account. checkTE returns the minimal value of $\Gamma/H(T)$ obtained after testing all possible subsets of particles in sector $n$. The particle assignment corresponding to the minimal value of $\Gamma/H(T)$ is printed on the screen. This value should be $\gg X_f$ to have chemical equilibrium; when this condition is satisfied, the correction to the abundance calculated *assuming* chemical equilibrium is approximately $\Delta Y/Y \approx X_f H/\Gamma$.

For the initialisation of the MICROMEGAS settings, one has to call the

• sortOddParticles(outText) command. It calculates all constrained model parameters, determines the number of sectors Ncdm, and fills the Ncdm+1 dimensional array McdmN containing minimal masses in each sector. McdmN[0]=0 corresponds to the SM sector.

• YdmNEq(T,$\alpha$) calculates the thermal equilibrium abundance at temperature T for particles of sector $\alpha$, where $\alpha$ has to be presented by a text label. For instance, YdmNeq(T,"1").

• vSigmaN(T,channel) calculates the thermally averaged cross-section $\langle v\sigma \rangle$ in [pb·c] units. Here channel is a text code specifying the reaction, e.g., vSigmaN(T,"1100") for $1, 1 \leftrightarrow 0, 0$ processes.

Note that to calculate $\langle v\sigma \rangle$ for processes with incoming bath particles, MICROMEGAS uses a short cut: to avoid calculating $\bar{n}_0$ for bath particles, it substitutes $\bar{n}_0 = s$ in $\langle v\sigma_{2010} \rangle$ in Eq. (20). To compensate for this factor, the rate of co-scattering processes (expressed in GeV)

is defined as

$$\Gamma_{2\to1} = \texttt{vSigmaN(T,"2010")}s(T)/3.894 \times 10^8. \tag{A.1}$$

To find the contribution of different processes to vSigmaN, one can call
• vSigmaNCh(T,channel,Beps,&vsPb) which returns an array of annihilation processes together with their relative contributions to the total annihilation cross-section. The cross-section is given by the return parameter vsPb in [pb·c] units. The elements of the array are sorted according to their weights, with the last element having weight=0. The structure of this array is identical to vSigmaTCh which was defined for one-DM models, see [49]. The input parameter channel is again written in text format. The memory allocated by outCh can be cleaned after usage with the command free(outCh). The following lines of code give an example for how to use this function:

```
aChannel*outCh=vSigmaNCh(T, "1100", Beps, &vsPb);
for(int n=0;;n++)
{ if(outCh[n].weight==0) break;
  printf(" %.2E  %s %s -> %s %s\n", outCh[n].weight,
  outCh[n].prtcl[0],outCh[n].prtcl[1],outCh[n].prtcl[2],
  outCh[n].prtcl[3]);
}
free(outCh);
```

• darkOmegaNTR(TR,Y,Beps,&err) solves the equation of the thermal evolution of abundances starting from the initial temperature TR and returns the total $\Omega h^2$ in Eq. (24). The array Y has to contain the initial abundances at the temperature TR. After completion, Y[k] contains the abundances of sector $k-1$ at the temperature Tend defined by the user.[5] The parameter TR is assigned to the global variable Tstart.

The error code err is a binary code which can signal several problems simultaneously. The codes 1, 2, 3, generated by the integration program simpson, mean

```
1 - NaN in integrand;
2 - too deep recursion;
3 - loss of precision;
```

In general, these codes can be treated as warnings. Nonetheless it can be useful to check the calculation of integrals, which lead to problems, using e.g. the gdb debugging tools. For more detail, see also the explanation of the simpson routine in the MICROMEGAS manual. The code 128 signals a problem in the solution of the differential equation, for example this problem can arise when TR is too large.
• darkOmegaN(Y,Beps,&err) calls darkOmegaNTR to solve the equations of the thermal evolution of abundances in the temperature interval [Tend,Tstart]. In each sector, the function looks for the temperature $T_i$ where $Y_i(T_i) \approx Y_{eq}(T_i)$. Tstart is then defined as the minimum value of $T_i$. If Tstart is not found, then the error code 64 is generated and darkOmegaN returns NaN.
• YdmN(T,$\alpha$) presents the evolution of abundances for particles of sector $\alpha$ calculated by darkOmegaN or darkOmegaNTR for T $\in$ [Tend,Tstart].

For the above functions, MICROMEGAS provides the possibility to selectively exclude part of the terms in the evolution equation. This is realised via the string ExcludedFor2DM which can be assigned specific keywords. The keyword "DMdecay" excludes decay processes which

---

[5]By default Tend = $10^{-3}$ GeV. However, when the decay contribution is important, it is preferable to choose a smaller value such as Tend = $10^{-8}$ GeV.

contribute to the DM evolution, while the keyword "1100" excludes $1,1 \leftrightarrow 0,0$ processes. To exclude co-scattering ($2,0 \leftrightarrow 1,0$ or $1,0 \leftrightarrow 2,0$ ) processes, set

    ExcludedFor2DM="2010";

The behaviour is reset to default by `ExcludedFor2DM=NULL;` which means that all channels are included.

An option to calculate the decay width of any particle including the contributions from channels with different numbers of outgoing particle is also provided:
• pWidthPref(particle_name, pref) defines the switches for pWidth, the function which calculates the tree-level width and decay branching ratios for a given particle. By default, pWidth checks the value of the useSLHAwidth flag, if useSLHAwidth!=0 and there are decay data in the loaded SLHA file, then pWidth returns the value stored in the file. Otherwise the widths are calculated at tree level including only channels with the minimal number of outgoing particles. pref allows to override this prescription for a single particle. It can take the values

0 – width is calculated using processes with minimal number of outgoing particles.

1 – width is calculated using processes with minimal and next to minimal number of outgoing particles excluding processes with s-channel resonances to avoid double counting.

2 – width is read from the SLHA file; if the SLHA file does not contain widths, it is calculated as in 0.

3 – width is read from the SLHA file; if the SLHA file does not contain widths, it is calculated as in 1.

4 – the default option of pWidth is used.

**Additional remarks:** In Eqs. (18) and (19) we have not included terms $\langle \sigma_{1000} v \rangle$ or $\langle \sigma_{2000} v \rangle$ since they do not appear in usual models where dark sector particles have a different discrete charge than SM particles.

Note also that, for the computation of co-scattering, the user defines which particles belong to sector 1 and sector 2. If a particle of dark sector 1 is wrongly assigned to dark sector 2 while it is in thermal equilibrium with sector 1 (for example if the singlet is in thermal equilibrium with the triplet), the two abundance equations will be solved and should give the same result as the single abundance equation, that is $Y(\text{heavier particles}) = 0$ and $Y(\text{lightest particle}) = Y$ of the single equation.

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
