# Peer review of "Co-scattering in micrOMEGAs: a case study for the singlet-triplet dark matter model"

_SciPost Physics, doi:SciPost Phys. 13, 124 (2022)_

## Round 1 · Referee Report · Anonymous · 2022-8-19

Strengths
1. Implements the co-scattering mechanism in micrOMEGAs, which has never been done in public tools before.
2. Very detailed
3. Well written
Report
This paper presents, for the first time, the incorporation of co-scattering processes in micrOMEGAs. The co-scattering mechanism can be the dominant early-universe production mechanism for DM in certain regions of parameter space, but had not been implemented in any public tools before. The authors study a fermionic singlet-triplet model as an example scenario, and describe the rich phenomenology of such a model. The singlet-triplet extension of the SM is outlined in section 2, where the authors introduce two new fermionic fields: a SM singlet and an electroweak triplet. The singlet-like state plays the role of the dark matter, but due to small mass splittings, the triplet-like state also plays a crucial role in the dynamics. The authors consider operators up to dimension-5, and list all the interactions of the new fields with the electroweak gauge fields and Higgs. In section 3, they go on to solve the Boltzmann equations while incorporating the co-scattering mechanism, and analyze the parameter space in section 4. The authors take specific interest in mapping the boundary region between where co-annihilation and co-scattering are the dominant mechanisms, and they show that there are new regions in the parameter space of the model associated with co-scattering. Additionally, they evaluate the LHC constraints on the long-lived charged states via disappearing charged track searches. Finally, the micrOMEGAs routines for co-scattering are outlined in appendix A.
The paper is written very well, and contains novel and important contributions to relic density calculations. I believe that the results and wider impact on relic density calculations deem it absolutely deserving to be published. I am happy to recommend it for publication.

---

## Editorial Decision

published